# Satisfaction and Demands of Indoor Space in the High-Density Residential Areas in the COVID-19 Era

**Jing Yang [1], Jiahang Xu [1,*] , Tingting Hu [2] and Jianing Cao [3,*]**

[1] School of Architecture, Southeast University, Nanjing 210000, China; 101010756@seu.edu.cn
[2] Nanjing Bangjian Practice Architectural Design Office Co., Ltd., Nanjing 210000, China; hutingting@bjuag.com
[3] Nanjing Yi Ju Construction Co., Ltd., Nanjing 210000, China
[*] Correspondence: jzxjh954@163.com (J.X.); pusherlly@163.com (J.C.)

**Abstract:** The last few years have witnessed a change in residents' demand for indoor space due to the COVID-19 pandemic. From the perspective of residential satisfaction in the urban areas in various levels of COVID-19 severity, the household survey was conducted to explore the changing residential demands. The IBM SPSS Statistics was employed to analyze the survey data with a focus on the relationship between pandemic severity and residents' satisfaction, as well as the future influence of COVID-19 on indoor space and the varying demands. Correlation analysis was performed. The variables included in the correlation analysis were the following: urban epidemic severity, number of confirmed cases, density of confirmed cases, regional pandemic severity and satisfaction of different indoor spaces. This study revealed that the hallway, bathroom, living room and master bedroom are key areas in which the residential demands are concentrated. These should be paid attention to in the future residential design.

**Keywords:** COVID-19; indoor space; satisfaction; varying demands; space strategy

## 1. Introduction

The COVID-19 pandemic has had profound impacts on the economy and social life at the global scale. As a fundamental of human society, residential building plays a crucial role in any strategies to deal with pandemic. Meanwhile, the COVID-19 pandemic has significant impacts on living spaces. For instance, it is a common measure to limit the movement people, which saw the restriction of access to outdoor spaces. As a result, the indoor space as well as the outdoor community space are being used more often than ever [1]. In addition, there is a growing level of importance for the living space in order to meet people's health and safety needs [2]. Therefore, there is a growing level of demand on the quality of living spaces [3]. This is particularly the case in high-density high-rise communities.

COVID-19 has changed the way of life and work, as well as the demands of the living environment [4]. As a result, working at home has been used as a common means to effectively contain the pandemic. Indeed, a new mode of "living + working" emerges in indoor space. Due to pandemic prevention measures such as disinfection and quarantine, the hallway (the entrance space) has to meet the new residential demands. These include the space of disinfection, changing clothes and the in-and-out flow of pollutants to be separated from the living space.

Therefore, this study places focus on the utilization status and the varying demands of indoor space in high-density high-rise residential areas, aiming to accurately identify critical issues associated with indoor spaces and improve people's quality of life in the post-COVID-19 era.

The structure of this paper is as follows: The second section reviews literature related to the relationship between pandemic and residential indoor space. In the third section,

the research methods are introduced in detail, including sample selection, questionnaire survey and data analysis methods. The research results are analyzed, and the related issues of indoor space in the context of COVID-19 are summarized in the next section. The final section presents conclusions and puts forward the design strategy of two types of indoor spaces in the COVID-19 era accordingly.

**2. Literature Review**

A large number of studies have been undertaken to examine the impacts of pandemic on the residential indoor space. For instance, some measures were put forward post the SARS pandemic in 2003. Lv [5] explored the types of activities in the residential areas and highlighted the importance of cultural and sports activities in the residential areas from the perspectives of development, management, design and planning. Qin [6] put forward related prevention and control measures in the architectural design of residential areas, pointing out that desirable natural ventilation and sufficient sunlight are the issues to be considered in the context of a pandemic. Dong et al. [7] highlighted a series of issues related to SARS, such as a lack of indoor communication space in residential areas, the health and safety issue of balconies and bathrooms, etc.

There are also a number of studies that focused on the impacts of COVID-19 on the indoor space of residential areas. The main focuses of these studies include the demands of indoor space, the design of indoor space, the indoor physical environment, indoor facilities and equipment, and so on:

1.  Pandemics and the demands of indoor space: Tleuken et al. [8] suggested that the demands and consumption of household water and energy increased during the COVID-19 pandemic; however, they were not significantly different from previous periods. Tokazhanov et al. [9] argued that health and safety is the most critical category of indicators for residential buildings, while innovative smart technologies play a crucial role in preventing the spread of the virus. In addition, the elderly or disabled people have urgent demands of home-based medical services [10].

2.  Pandemics and the design of indoor space: Healthy residence and adaptable indoor space have been paid a lot of attention since the COVID-19 pandemic. Xu [3] pointed out that improving the type of layout plays a crucial role in promoting the quality of indoor space and discussed adaptable indoor space from the perspectives of structure and layout. Lang [11] and Li et al. [12] analyzed strategies of indoor space design to create healthy residences and allow for better adaptability. For example, indoor space can be adapted to different family structures by combining functional compound spaces with adaptable spaces.

3.  Pandemics and the indoor physical environment: The physical environment not only affects spatial comfort but is also the key to residents' mental health and pandemic prevention. Lepore et al. [13] performed regression analysis to assess the indoor ventilation measures and discovered that existing ventilation measures are insufficient to satisfy the demands of ventilation safety during the pandemic. A new type of wall material is used to adjust indoor humidity, and the relative air humidity in the range of 40~60% is more conducive to preventing the spread of COVID-19 [14]. Therefore, the regulation of air humidity is extremely important in terms of indoor design.

4.  Pandemics and indoor facilities and equipment: Lu et al. [4] highlighted changes in living spaces and daily life facilitated by new technologies in the context of COVID-19. For example, the combination of telehealth and healthy residence plays a critical role in dealing with the pandemic [15]. The ultraviolet germicidal irradiation, bipolar ionization, vertical gardening, and indoor plants also provide key technical support for the design transformation of healthy residence in the post-COVID-19 era [16].

In addition, satisfaction is the most intuitive reflection of residents' demands. Similarly, it is an essential element for data support and quantitative analysis. There are also extensive studies on the residential satisfaction. Galster et al. [17] introduced 'marginal residential improvement priority', a new housing indicator, and highlighted the significant association

between the satisfaction and residential environment, house perception, and neighborhood relationship. KamacI-Karahan et al. [18] conducted correlation and regression analysis to examine the changes of residents' satisfaction after the disaster with beneficiary status, past residential experiences, the social environment, and socio-demographic factors as controlling variables. An independent sample *t*-test was used to assess the evaluations of residents about the settings before and after the disaster. Salamone et al. [2] and Haverinen-Shaughnessy et al. [19] investigated the correlation between satisfaction and indoor environmental quality using a descriptive approach, predictive models and statistical analysis. Their study showed that visual comfort and energy retrofitting can change residents' satisfaction. However, very few studies examined the indoor space of residential building during the pandemic from the perspective of residential satisfaction.

Meanwhile, the existing literature on assessment methods of the COVID-19 pandemic mostly focused on the impact on housing market demand, rather than the demand for different indoor spaces. Batalha et al. [20] employed a parish-level treatment to compare the rental market of different civil parish in Lisbon, Portugal, which provides a mode to compare different regions as research samples. Liu et al. [21] pointed out that the pandemic had shifted housing demand from high to low population density areas, based on various housing indicators in the U.S. housing market. According to a stepwise mediation effect test, Li et al. [22] found that the impact of the pandemic on housing market demand is different between China and western countries.

In summary, there have been extensive studies on the impact of the COVID-19 pandemic on indoor space. However, very few studies focused on the differences in satisfaction and demand of various indoor spaces in different regions. Therefore, this study sought to compare changes in satisfaction and demands on the same issues in indoor spaces with different levels of COVID-19 severity, so that the critical issues associated with various indoor spaces could be identified accurately, which can lead to better design strategies.

## 3. Research Methods

Selected residential areas were drawn from Chinese cities with different levels of COVID-19 severity. Household surveys were conducted to examine the changes in users' demands in indoor space. With the assistance of the residential manager, we were able to approach the households of the users who volunteered to complete the questionnaire survey and interviews. Corresponding strategies could then be developed to improve the indoor space design.

By October 2020, the first wave of COVID-19 in China had moderated. In that case, the sample of this study was classified according to the COVID-19 severity in Chinese cities before October 2020. Five residential areas were drawn from three cities in the Yangtze River Delta that were classified with low levels of COVID-19 severity, i.e., Nanjing, Wuxi, and Hangzhou. These five residential areas are Huagang Hong Town in Nanjing, Dahua Jinxiu Huacheng Yuejiangshan in Nanjing, Wukuang Chongwen Golden Town in Nanjing, Yulan West Garden Phase II in Wuxi, and Houchao Mansion in Hangzhou.

Six residential areas were drawn from Wuhan city, which was classified with high level of COVID-19 severity. These six residential areas are: Paris Spring in Jiangxia District, Shimaolin Yu'an in Jiangxia District, Liantou Longwan Phase III in Jiangxia District, East Lake Chutianfu in Hongshan District, Luojiayayuan in Hongshan District, and Dahua South Lake Park Mansion Phase III in Hongshan District.

In other words, the sample was drawn from 11 residential areas in 4 cities. They are representative high-density residential areas with rigid demand and were renovated recently. Building areas ranged from 70 to 140 square meters.

The sample size of the questionnaire survey was determined according to established principles [23]. The sample size should be 5–10 times of the number of scale-based questions. There are 46 questions in the survey of this study; therefore, the sample size should be 230 to 460. A total of 327 valid responses were received, which satisfied the sample size requirement. There are 150 effective household-survey samples in the cities with a low

level of COVID-19 pandemic severity and 177 effective household-survey samples in the cities with a high level of COVID-19 pandemic severity. The number of effective samples in each residential area ranges from 27 to 33, which is rather balanced.

Meanwhile, the online questionnaire survey was conducted, and random sampling was adopted. A total of 1698 questionnaires were collected, 86.4% of which are valid. Respondents included residents from cities with severe pandemic such as Hubei Province, as well as those cities with moderate pandemic such as North and Central China. The online questionnaire explores the residents' living habits, current problems, and future demands during the COVID-19 period, so as to give feedback on the general demands of domestic residents.

Five-point Likert scale was used to measure the respondents' opinion to each question. The Likert scale ranges from 1 to 5: 1 represents "strongly disagree", 2 represents "disagree", 3 represents "generally agree", 4 represents "agree", and 5 represents "strongly agree". Satisfaction was quantified based on questionnaire scores.

IBM SPSS Statistics was used to analyze the correlation between residents' satisfaction and pandemic severity. A series of variables were used to measure the pandemic severity, such as "Number of Confirmed Cases", "Density of Confirmed Cases", "Urban Epidemic Severity", and "Regional Epidemic Severity". Among them, the first two variables were used to accurately and intuitively reflect the COVID-19 pandemic severity. The latter two variables were used to represent the severity of the sample location at the macro-level. On the other hand, the setting of multiple variables was used to verify the robustness of the data. Furthermore, One-way analysis of variance was employed to analyze the specific change trend of satisfaction.

## 4. Results and Discussion

### 4.1. The Household Surveys on Residential Satisfaction

In total, 150 responses were received from cities with a low level of COVID-19 pandemic severity. These residents provided 924 entries on various indoor problems. These entries are mainly related to bathrooms (19.8%), followed by master bedrooms (14.8%), kitchens (14.2%), balconies (12.3%), hallways (9.3%), living rooms (9.1%), secondary bedrooms (9.0%), dining rooms (8.4%), and studies (3.0%). Most of these dissatisfactions concentrated on bathrooms, master bedrooms, kitchens, and balconies (Figure 1).

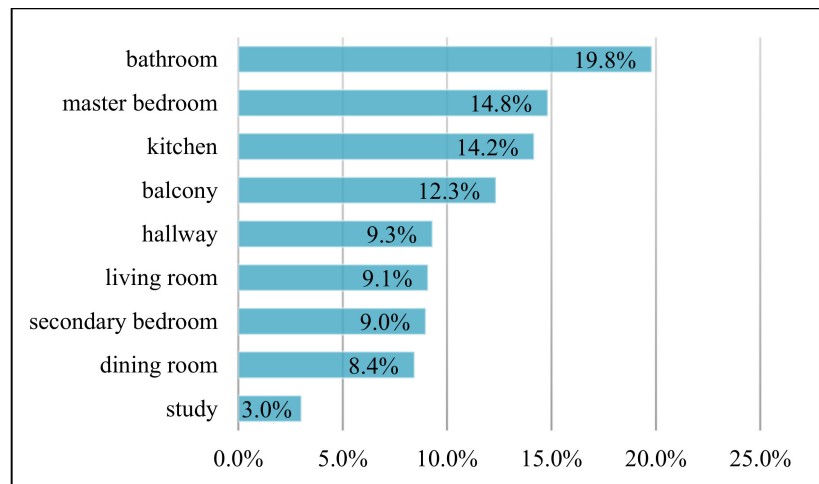

**Figure 1.** Indoor-space-related issues in cities with low level of COVID-19 pandemic severity.

In total, 177 responses were received from cities with a high level of COVID-19 pandemic severity. These residents provided 627 entries on various indoor space problems. The entries are mainly related to hallways (16.9%), followed by the kitchen (15.6%), bathroom (13.2%), living room (12.4%), dining room (12.0%), secondary bedroom (10.5%), master

bedroom (8.3%), balcony (8.3%), and study (2.7%). It can be observed that cities with high levels of COVID-19 pandemic severity suffer from an ever-increasing residential dissatisfaction in hallways, living rooms, and dining rooms. In contrast, the entries related to the bathroom, master bedroom, and balcony decreased significantly. In addition, the entries related to the kitchen, secondary bedroom, and study room remain unchanged. Therefore, the residents in cities with a high level of COVID-19 pandemic severity have greater demands of space quality for hallways, living rooms, and restaurants. The specific correlation between satisfaction with indoor space and urban pandemic severity will be further explored in the following section (Figure 2).

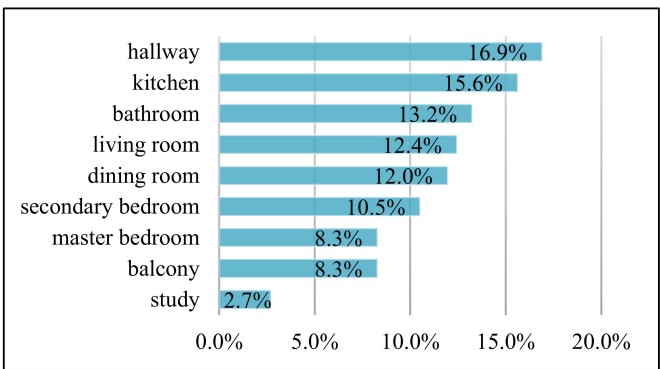

**Figure 2.** Indoor-space-related issues in cities with a high level of COVID-19 pandemic severity.

The characteristics of respondents may have a potential influence on the questionnaire survey results. For instance, age, occupation, and income are potential factors that may affect residents' satisfaction with different indoor spaces. Satisfaction with living condition may decrease with age and income [24]. Unfortunately, most respondents were reluctant to disclose such private information. However, selected residential areas with similar characteristics could generally represent the same category of respondents.

*4.2. The Online Questionnaire about the High-Density Residential Areas All over the Country*

4.2.1. The Varying Demands of Living Space in the Post-COVID-19 Era

People staying at home for longer period of time during the pandemic have more demands for indoor space. As a result, potential issues of many property types are fully exposed. Indeed, health, hygiene, and user experience become the key aspects of indoor space. The residential demands of personal and kitchen hygiene increase significantly. The demand for eating at home is ranked in second place. The demands of home-based office space and living room for family gathering and entertainment become strong. Meanwhile, there is a growing level of demand on establishing a fitness space at home (Figure 3).

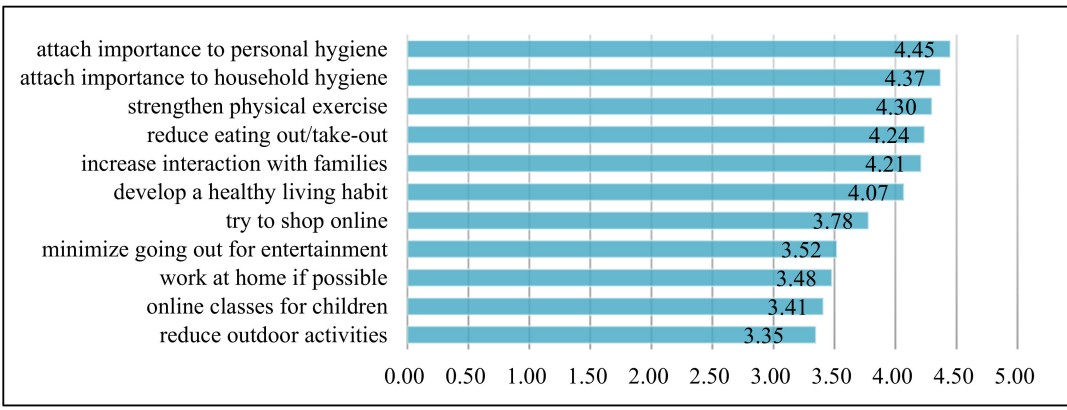

**Figure 3.** Changes in the people's living habits in the COVID-19 era.

4.2.2. The Varying Demands of Facilities and Equipment in the Post-COVID-19 Era

In the future, the indoor-space facilities and equipment will be more intelligent and convenient, which serve as the added value of the living space. After the COVID-19 outbreak, the residential demands of intelligent facilities and equipment will become more profound and accurate. More attention will be paid to smog removal by fresh air, water-purification equipment, and no-contact facilities (Figure 4).

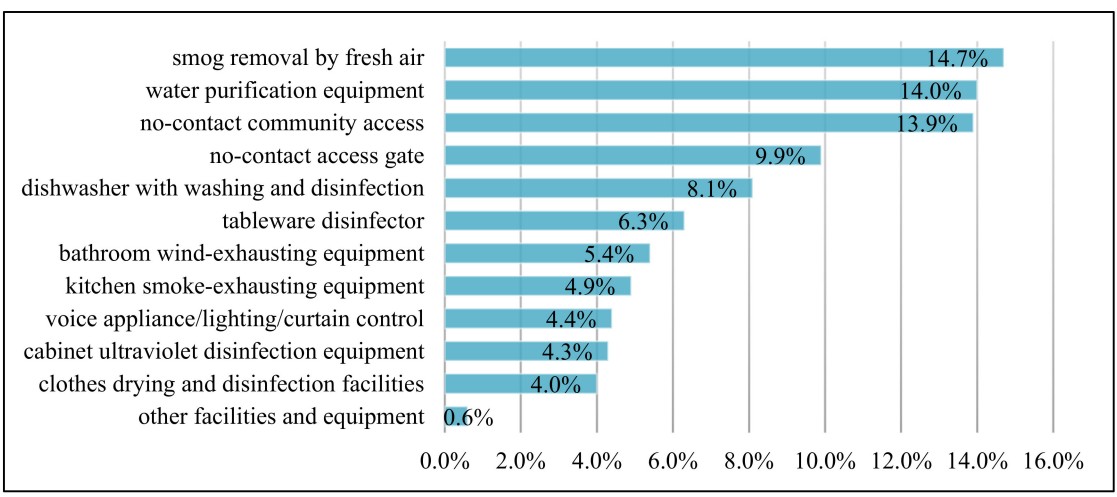

**Figure 4.** Residents' major concerns in the future.

4.2.3. The Changing Trend of Housing Types in the Post-COVID-19 Era

In terms of optimization of indoor space after the pandemic, 70.0% of the residents expressed that the internal partition walls of their apartment can be removed or moved in order to have a more flexible space. Moreover, 59.7% of the residents would like to have a two-suite living space, which is conducive to having no interference between the two generations and quarantine during the pandemic. Furthermore, 63.4% of the respondents express willingness to set up an independent housekeeping space; 81.5% of the residents expressed they would enjoy dry-and-wet separation in the public bathrooms; And 56.4% of the respondents would like set up two balconies in the north and south, as well as a separation between leisure and drying (Figure 5).

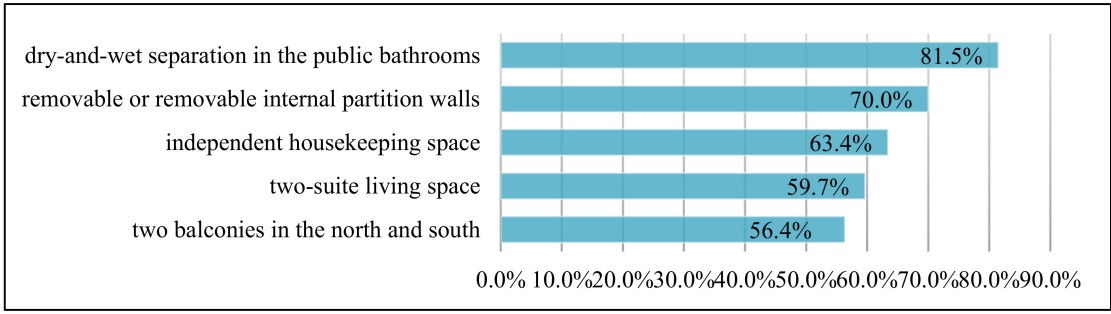

**Figure 5.** Optimization of indoor space.

*4.3. The Correlation between Pandemic Severity and Residents' Satisfaction*

Pandemic severity varies in different regions of the world in the context of COVID-19. Various indicators were employed in this study to measure the pandemic severity. "Urban Epidemic Severity" is defined at a macro-level. "Number of Confirmed Cases" refers to the total number of confirmed cases in the administrative district of the residential area released by the Municipal Health Commission by October 2020. These include 10 in Qixia District of Nanjing, 6 in Pukou District of Nanjing, 13 in Jianye District of Nanjing, 7 in Binhu

District of Wuxi, 12 in Shangcheng District of Hangzhou, 860 in Jiangxia District of Wuhan, and 4718 in Hongshan District of Wuhan, indicating the difference in the residential areas with the similar "Urban Epidemic Severity". "Density of Confirmed Cases" refers to the "Number of Confirmed Cases" as a proportion of the total population of each administrative district, so as to represent pandemic severity of different cities or regions in an accurate manner. The density is 0.01‰ in Qixia District of Nanjing, 0.01‰ in Pukou District of Nanjing, 0.02‰ in Jianye District of Nanjing, 0.01‰ in Binhu District of Wuxi, 0.01‰ in Shangcheng District of Hangzhou, 0.88‰ in Jiangxia District of Wuhan, and 2.73‰ in Hongshan District of Wuhan, respectively (Table 1).

**Table 1.** The indicators of pandemic severity.

| Indicator | Urban Pandemic Severity | Number of Confirmed Cases | Density of Confirmed Cases | Regional Pandemic Severity |
|---|---|---|---|---|
| Huagang Hongfu City (Qixia District of Nanjing) | non-key | 10 | 0.01‰ | low |
| Dahua Jinxiu Huacheng Yuejiangshan (Pukou District of Nanjing) | non-key | 6 | 0.01‰ | low |
| Wukuang Chongwen Golden City (Jianye District of Nanjing) | non-key | 13 | 0.02‰ | low |
| Yulan West Garden Phase II (Binhu District of Wuxi) | non-key | 7 | 0.01‰ | low |
| Houchao Mansion (Shangcheng District of Hangzhou) | non-key | 12 | 0.01‰ | low |
| Paris Spring (Jiangxia District of Wuhan) | key | 860 | 0.88‰ | medium |
| Shimaolin Yu'an (Jiangxia District of Wuhan) | key | 860 | 0.88‰ | medium |
| Liantou Longwan Phase III (Jiangxia District of Wuhan) | key | 860 | 0.88‰ | medium |
| East Lake Chutianfu (Hongshan District of Wuhan) | key | 4718 | 2.73‰ | high |
| Luojiayayuan (Hongshan District of Wuhan) | key | 4718 | 2.73‰ | high |
| Dahua South Lake Park Mansion Phase III (Hongshan District of Wuhan) | key | 4718 | 2.73‰ | high |

Note: "Number of Confirmed Cases" refers to the related data of each city by October 2020. "Density of Confirmed Cases" refers to the proportion of "Number of Confirmed Cases" in the total population of each administrative district. Source: Official data released by the municipal governments.

Correlation analysis results showed that the satisfaction of bathroom, master bedroom, living room, and study are correlated to "Urban Epidemic Severity". The hallway-based data show significant correlation, while the data on the other spaces show no significant correlation. The data on hallway and living room show significant correlation with "Number of Confirmed Cases" and "Density of Confirmed Cases"; the data on bathroom, master bedroom, and study show less significant correlation. Therefore, the selection of indicators to measure pandemic severity has certain impacts on the correlation analysis results of residential satisfaction of some indoor spaces (Table 2).

**Table 2.** The correlation results of indoor space satisfaction.

| Sig. | Bathroom | Master Bedroom | Kitchen | Balcony | Hallway | Living Room | Secondary Bedroom | Dining Room | Study |
|---|---|---|---|---|---|---|---|---|---|
| Urban Pandemic Severity | 0.027 * | 0.013 * | 0.259 | 0.738 | 0.000 ** | 0.046 * | 0.214 | 0.228 | 0.012 * |
| Number of Confirmed Cases | 0.088 | 0.116 | 0.084 | 0.220 | 0.028 * | 0.007 ** | 0.286 | 0.826 | 0.255 |
| Density of Confirmed Cases | 0.056 | 0.066 | 0.089 | 0.271 | 0.008 ** | 0.006 ** | 0.246 | 0.672 | 0.149 |
| Regional Pandemic Severity | 0.034 * | 0.032 * | 0.110 | 0.367 | 0.001 ** | 0.007 ** | 0.212 | 0.494 | 0.068 |

Note: The Sig. value represents significance; * represents significant correlation at the level of 0.05 (double-tailed); ** represents significant correlation at the level of 0.01 (double-tailed).

"Number of Confirmed Cases" and "Density of Confirmed Cases" varied significantly between cities with low level of COVID-19 severity and those with high level of COVID-19 severity. In addition, there is a large gap between Jiangxia District and Hongshan District of Wuhan. Therefore, the correlation results could be biased. Therefore, the indicator of

"Regional Epidemic Severity" is introduced to classify the sample into three levels of low, medium, and high, according to "Number of Confirmed Cases" and "Density of Confirmed Cases". Consequently, further correlation analysis was conducted on the satisfaction with indoor space. The results show that the data on the bathroom and master bedroom have some correlation with "Regional Pandemic Severity", while the data on the hallway and living room show significant correlation with "Regional Pandemic Severity". These results are similar to the correlation results of "Urban Epidemic Severity", and the criterion of "Regional Pandemic Severity" is more suitable for this study (Table 2).

*4.4. Changing Trend of Indoor Space Satisfaction*

In order to further explore the specific changes of indoor space satisfaction among three levels of low, medium, and high, the non-significant correlation items in "Regional Pandemic Severity" are eliminated. As for the satisfaction of the remaining functional space, the one-way analysis of variance was conducted, and the results of the three levels are compared between pairs (Table 3).

**Table 3.** The one-way analysis of variance of "Regional Epidemic Severity".

| Sig. | | Bathroom | Master Bedroom | Kitchen | Balcony | Hallway | Living Room | Secondary Bedroom | Dining Room | Study |
|------|--------|----------|----------------|---------|---------|----------|-------------|-------------------|-------------|-------|
| low | medium | 0.102 | 0.042 * | - | - | 0.000 ** | 0.312 | - | - | - |
| | high | 0.050 * | 0.042 * | - | - | 0.000 ** | 0.009 ** | - | - | - |
| medium | low | 0.102 | 0.042 * | - | - | 0.000 ** | 0.312 | - | - | - |
| | high | 0.691 | 1.000 | - | - | 1.000 | 0.072 | - | - | - |
| high | low | 0.050 * | 0.042 * | - | - | 0.000 ** | 0.009 ** | - | - | - |
| | medium | 0.691 | 1.000 | - | - | 1.000 | 0.072 | - | - | - |

Note: The Sig. value represents significance; * represents significant correlation at the level of 0.05 (double-tailed); ** represents significant correlation at the level of 0.01 (double-tailed).

"Number of Confirmed Cases" and "Density of Confirmed Cases" varied significantly between cities with low level of COVID-19 severity and those with high level of COVID-19 severity. In addition, there is a large gap between Jiangxia District and Hongshan District of Wuhan. Therefore, the correlation results could be biased. Therefore, the indicator of "Regional Epidemic Severity" is introduced to classify the sample into three levels of low, medium, and high, according to the "Number of Confirmed Cases" and the "Density of Confirmed Cases". Consequently, further correlation analysis was conducted on satisfaction of indoor space. The results show that the data on the bathroom and master bedroom have some correlation with "Regional Pandemic Severity", while the data on the hallway and living room show significant correlation with "Regional Pandemic Severity". These results are similar to the correlation results of "Urban Epidemic Severity", and the criterion of "Regional Pandemic Severity" is more suitable for this study (Table 2).

*4.5. Issues Related to Indoor Space in the Context of COVID-19*

All issues related to indoor space are classified into four categories, i.e., environmental quality, space function, space relationship, and facility design, according to the survey data. Each category has several items according to different functions. The data of the key functional space is sorted out. The bathroom, living room, master bedroom, and hallway are closely related to the pandemic. Therefore, the analysis of the issues related to indoor space mainly involves these four spaces.

Based on the survey data from cities with a low level of COVID-19 pandemic severity, issues related to different indoor spaces with high correlation with the pandemic were

selected. For the bathroom-related issues, poor natural ventilation and lighting are the most prominent, which lead to both inadequate air circulation and increasing risk of virus spread. There are many other issues such as insufficient storing space and bathroom facing the kitchen door. These issues result in low efficiency in the residents' hang-washing and disinfection on one hand and increasing risk of the pandemic on the other hand. However, these entries account for less than 10% of the total, meaning that such issues as the size of bathroom and entrance location should be considered as secondary factors to improve indoor spaces in cities with low levels of COVID-19 pandemic severity (Table 4).

In addition to the bathroom, the main problems of other indoor spaces are poor ventilation and lighting, small activity space, insufficient storage space, and so on. These problems are closely related to the pandemic. Ventilation and lighting not only affect air circulation but also reduce residents' willingness to use indoor spaces. The small activity space is unable to meet the demands of residents during the household quarantine. For example, working at home has become a new normal at present. Insufficient storing space cannot meet the demands of temporary storage of take-away items, emergency-food storage, as well as the installation and storage of appliances and other equipment. In addition, the demand of separate living and service balconies is related to the pandemic. Daily leisure activities and laundry-and-drying need to be differentiated during the household quarantine, which reduces the risk of virus spread to a certain degree. However, such entries account for merely 3.5%, indicating that the related demands in the residential areas are not regarded as the main issues in cities with low levels of COVID-19 pandemic severity.

Based on the survey data from cities with a high level of COVID-19 pandemic severity, it can be observed that the hallway-related entries are the most prominent. Among them, the entries related to insufficient storing space accounted for the largest proportion. There is no disinfection equipment or pollutant recovery area, thus reducing the efficiency of household disinfection measures such as hand washing and item disinfection on one hand and failing to effectively isolate external pollution on the other hand. The hallway is an important area to separate internal and external space. Therefore, the problem of no separation between the hallway and living room also affects the effectiveness of hallway-based pandemic prevention.

Data were compared between cities with low levels of COVID-19 pandemic severity and those with high levels of COVID-19 pandemic severity. It can be observed that the residents have the highest demand of the hallway-based storing space. Meanwhile, the demand of hallway-based ventilation and lighting increases significantly in cities with a high level of COVID-19 pandemic severity. Furthermore, there are new demands of internal and external separation, disinfection equipment, and pollutant recovery area. Therefore, pandemic severity not only affects the residents' demand of indoor space but also gives rise to new demands of some indoor spaces (Table 4).

The proportion of the bathroom-related entries from cities with high levels of COVID-19 pandemic severity is relatively lower than that from cities with low levels of COVID-19 pandemic severity, though still attracting wide public concern. Among them, there are many entries related to lighting and ventilation, while the proportion of the entries related to small spaces and insufficient storing space increased significantly. In addition, there are new entries, such as no dry-and-wet separation and floor drains prone to water accumulation. The long-term retention of humid air in the bathroom under poor ventilation conditions leads to increasing risk of virus spread and virus accumulation. The entries related to dry-and-wet separation in cities with high levels of COVID-19 pandemic severity accounted for the largest proportion. By contrast, these are not serious issues in from with low level of COVID-19 pandemic severity, thus indicating that the bathroom-based dry-and-wet separation plays an important role in terms of pandemic prevention. The entries related to other spaces are similar to those in cities with low levels of COVID-19 pandemic severity. These include poor ventilation and lighting, small activity space, and insufficient storage space, therefore reflecting the common residential demands in most cities in China.

**Table 4.** The proportion of the entries in the context of COVID-19.

| | Space Category | Environment Quality | | Space Relationship | Space Function | | Facility Design |
|---|---|---|---|---|---|---|---|
| **bathroom** | problem description | poor ventilation and lighting | | the bathroom facing the kitchen door | insufficient storing space | no dry-and-wet separation | slow drainage of the floor drainžžand prone to water accumulation |
| | Cities with low level of COVID-19 pandemic severity | 56.2% | | 3.3% | 8.2% | - | - |
| | Cities with high level of COVID-19 pandemic severity | 26.5% | | - | 14.5% | 30.1% | 3.6% |
| **master bedroom** | problem description | poor ventilation and lighting | | no | insufficient storage space | small space | no |
| | Cities with low level of COVID-19 pandemic severity | 7.3% | | - | 23.4% | 27.0% | - |
| | Cities with high level of COVID-19 pandemic severity | 3.8% | | - | 36.5% | 13.5% | - |
| **kitchen** | problem description | poor ventilation and lighting | | no | insufficient storage space | small space | smoke drainage and other pipe problems |
| | Cities with low level of COVID-19 pandemic severity | 15.2% | | - | 19.8% | 29.8% | - |
| | Cities with high level of COVID-19 pandemic severity | 27.6% | | - | 15.3% | 40.8% | 12.2% |
| **balcony** | problem description | insufficient lighting | poor ventilation caused by window opening | no separation between the living balcony and the service balcony | inconvenient drying caused byžžsmall depth | small space | inconvenience caused byžžthe way of door opening |
| | Cities with low level of COVID-19 pandemic severity | 30.7% | 2.6% | 3.5% | 4.4% | 50.0% | 1.8% |
| | Cities with high level of COVID-19 pandemic severity | - | 5.8% | 5.8% | 5.8% | 63.5% | - |
| **hallway** | problem description | poor ventilation and lighting | | no separation between the hallway and the living room | insufficient storage space | no hallway | no disinfection equipment or pollutant recovery area |
| | Cities with low level of COVID-19 pandemic severity | 17.5% | | - | 52.3% | 25.6% | - |
| | Cities with high level of COVID-19 pandemic severity | 27.3% | | 12.3% | 30.2% | - | 14.2% |

The proportion of the feedback in the COVID-19 context.

**Table 4.** *Cont.*

| | Space Category | Environment Quality | Space Relationship | Space Function | | Facility Design |
|---|---|---|---|---|---|---|
| **living room** | problem description | poor ventilation and lighting | no | restricted activity space | no separation from the dining room | no |
| | Cities with low level of COVID-19 pandemic severity | 44.0% | - | 40.5% | - | - |
| | Cities with high level of COVID-19 pandemic severity | 33.3% | - | 41.0% | 2.6% | - |
| **secondary bedroom** | problem description | poor ventilation and lighting | no | insufficient storage space | small space | no |
| | Cities with low level of COVID-19 pandemic severity | 17.9% | - | 5.3% | 47.4% | - |
| | Cities with high level of COVID-19 pandemic severity | 31.8% | - | 7.6% | 51.5% | - |
| **dining room** | problem description | poor ventilation and lighting | too close to the bathroom | insufficient storage space | small space | no |
| | Cities with low level of COVID-19 pandemic severity | 25.0% | - | - | 57.3% | - |
| | Cities with high level of COVID-19 pandemic severity | 20.0% | 2.7% | 1.3% | 42.7% | - |
| **study** | problem description | poor ventilation and lighting | no | small space | | no |
| | Cities with low level of COVID-19 pandemic severity | 63.6% | - | 22.7 | | - |
| | Cities with high level of COVID-19 pandemic severity | 31.8% | - | 51.5% | | - |

The proportion of the feedback in the COVID-19 context.

## 5. Conclusions

The current studies on the pandemic and indoor space are mainly based on the analysis of the demands for daily life, physical environment, facilities and equipment, and so on. However, little effort was made in terms of exploring the correlation between the varying demands of different functional space and the pandemic. In this study, by comparing the difference in satisfaction of indoor space with different levels of pandemic severity, the varying demands of residents in the context of COVID-19 are examined. Consequently, the influences of COVID-19 severity on various indoor spaces were examined. The hallway, bathroom, living room, and master bedroom are the indoor spaces on which the residential demands of are relatively concentrated. Therefore, focus should be placed on these aspects during the future design of indoor space, especially in high-rise high-density settings. Furthermore, the pandemic-related demands obtained from the residents' entries mainly focus on environmental quality, space function, space relationship, and equipment and facilities of indoor space.

This study revealed that two types of indoor spaces are highly correlated with changes in the COVID-19 severity. On one hand, it is the space with ever-growing demands of residents as the pandemic intensifies, for example, the living room. After the pandemic intensifies, residents have more demands of living room, and their indoor activity space is transformed into such a cohesive space as the living room. In this aspect, the optimal design of space relationship and space function should be carried out according to the varying demands of residents in the long-term household life during the pandemic. On the other hand, it is the space with steady demands of residents as the pandemic intensifies. In this aspect, more attention should be paid to the demands for epidemic prevention and temporary compound utilization. By sorting out the problems related to indoor space during the pandemic, this study revealed that different types of space have different demands for environmental quality, space function, space relationship, and facility design. The results of this study can provide a reference for designers in the interior design process of high-density residential areas in the COVID-19 era. Therefore, in terms of housing design, space or facilities should be subject to targeted design based on the specific demands.

The findings of this study are based on samples of high-density residential areas in different regions of China. Therefore, differences in satisfaction and demand for indoor spaces in the context of COVID-19 may vary according to level of density. Housing renovation may be the new trend in the future, which has become part of the response to COVID-19 in some countries. Consequently, on top of high-rise and high-density residential areas, future studies could be undertaken to examine the indoor spaces of old residential areas to be renovated.

**Author Contributions:** Author Contributions: Conceptualization, J.Y., J.X. and J.C.; methodology, J.X.; software, J.X.; validation, J.Y., J.X. and T.H.; formal analysis, J.X.; investigation, J.X., T.H. and J.C.; resources, J.C.; data curation, J.X. and T.H.; writing—original draft preparation, J.X.; writing—review and editing, J.Y., J.X., T.H. and J.C.; visualization, J.X.; supervision, J.Y.; project administration, J.Y. and J.C.; funding acquisition, J.Y. All authors have read and agreed to the published version of the manuscript.

**Funding:** This research received no external funding.

**Data Availability Statement:** Data available in a publicly accessible repository that does not issue DOIs Publicly available datasets were analyzed in this study. This data can be found here: [https://pan.baidu.com/s/1vpBAZV0FxK6lh8JA4TnvBQ?pwd=z0zb (accessed on 17 April 2022)].

**Conflicts of Interest:** The authors declare no conflict of interest.

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
