# Peer review of "Satisfaction and Demands of Indoor Space in the High-Density Residential Areas in the COVID-19 Era"

_buildings, doi:10.3390/buildings12050660_

Round 1
Reviewer 1 Report
Overall, this is an interesting study. However, some flaws affect the quality of the manuscript. In the following, I discuss the strengths and weaknesses of this paper.
Abstract:
The abstract needs to provide some information about the data analysis and the variables.
Introduction:
Overall, the introduction and literature review needs significant improvement.
The introduction needs some citations: for example, the sentence “As a result, the indoor space as well as the outdoor community space are used more often than ever” needs to be supported by other studies.
Literature review: the classification of the studies about indoor spaces of residential buildings seems interesting, however, the authors need to report more details about the findings of those studies (particularly, category 2 and the last two paragraphs of the literature review).
The transition from the introduction and literature review to the research gap is not reasonable. I would suggest adding one paragraph on how the lack of research on this topic drives the research goal and then discussing research goals and objectives.
Methodology:
This section needs to be improved significantly. The data collection has been discussed but the method they used to reach out to potential respondents is still vague. The authors need to provide some information about this. Besides, the data analysis method needs to be discussed in detail. Moreover, research variables are not clearly defined and the reader cannot understand where they are coming from and why the authors decided to use these variables.
Results and findings,
The results and findings of this research provide good information and can potentially add to the existing knowledge about the satisfaction of residents with the indoor spaces of residential buildings in the post-COVID era. I believe this part is the strongest part of the paper. The findings are discussed with a good focus on the different aspects of residential satisfaction and the discussion made on the “issues related to indoor spaces…” is interesting.
Conclusion:
The findings and the contribution of the studies have been discussed in this section. However, some important parts of the conclusion are missing including research limitations and recommendations for future research.
Reviewer 2 Report
This is an interesting study, comparing the difference in satisfaction of indoor space with different levels of pandemic severity, and examining the varying demands of residents in the context of COVID-19. The approach presented and the findings from this study would offer some new insight to people working in the associated field. The authors should change the title of the section 3 (Results) to ‘Results and Discussions’ instead. Otherwise, I do not have any further comments or suggestions for improvement of this article.
Reviewer 3 Report
The topic presented in the paper is current and interesting, the paper is well articulated and written, so minor revision are required before the publication.
In the Introduction Section the reference context should be expanded and the aim of the research should be better described. In addition, a paragraph describing the content of the sections of the paper should be added at the end.
In section 2 a focus on the international literature reference concerning the assessment methodology for the evaluation of the impacts of the COVID-19 pandemic on the housing market demand (e.g. see the Italian context) is missing.
In Section 3 Research Methods, it is advisable to deepen the description methodological approach and in particular of the survey, like the dissemination of the questionnaire, the period of reference etc.. It is also important to point out how the data collected have been processed to test their statistical robustness.
In Section 3 Results (there is a typo, it should be section 4), it is advisable to add a brief reflection on how the characteristics of the respondents potentially influenced/deviated the results obtained through the questionnaire. Furthermore, it is advisable to make the histogram labels in the figures more readable.
In Section 4 Discussion it is advisable to explain what decision-making processes the research carried out could support and what spin-offs it can have in terms of interior space design.
Round 2
Reviewer 1 Report
The edits made by the authors have improved the manuscript significantly and I believe the manuscript is in a good shape to be considered for publication after making all the edits.